# Data Mining and Deep Learning for Predicting the Displacement of “Step-like” Landslides

**DOI:** 10.3390/s22020481

**Published:** 2022-01-09

**Authors:** Fasheng Miao, Xiaoxu Xie, Yiping Wu, Fancheng Zhao

**Affiliations:** 1Faculty of Engineering, China University of Geosciences, Wuhan 430074, China; fsmiao@cug.edu.cn (F.M.); xiexx@cug.edu.cn (X.X.); zhaofancheng@cug.edu.cn (F.Z.); 2Engineering Research Center of Rock-Soil Drilling & Excavation and Protection, Ministry of Education, Wuhan 430074, China

**Keywords:** Three Gorges Reservoir, Baishuihe landslide, data mining, displacement prediction, VMD-FOA-BPNN

## Abstract

Landslide displacement prediction is one of the unsolved challenges in the field of geological hazards, especially in reservoir areas. Affected by rainfall and cyclic fluctuations in reservoir water levels, a large number of landslide disasters have developed in the Three Gorges Reservoir Area. In this article, the Baishuihe landslide was taken as the research object. Firstly, based on time series theory, the landslide displacement was decomposed into three parts (trend term, periodic term, and random term) by Variational Mode Decomposition (VMD). Next, the landslide was divided into three deformation states according to the deformation rate. A data mining algorithm was introduced for selecting the triggering factors of periodic displacement, and the Fruit Fly Optimization Algorithm–Back Propagation Neural Network (FOA-BPNN) was applied to the training and prediction of periodic and random displacements. The results show that the displacement monitoring curve of the Baishuihe landslide has a “step-like” trend. Using VMD to decompose the displacement of a landslide can indicate the triggering factors, which has clear physical significance. In the proposed model, the R^2^ values between the measured and predicted displacements of ZG118 and XD01 were 0.977 and 0.978 respectively. Compared with previous studies, the prediction model proposed in this article not only ensures the calculation efficiency but also further improves the accuracy of the prediction results, which could provide guidance for the prediction and prevention of geological disasters.

## 1. Introduction

Landslides occur frequently around the world and are one of the most destructive geological disasters in the world [1,2]. Landslide displacement prediction is one of the geological engineering problems that at present has not been solved, especially for mountain and reservoir areas. Reservoir impoundment usually affects the surrounding geological environment, resulting in landslide disasters. As the largest power station in terms of installed capacity in the world since 2012, the water level of the Three Gorges Reservoir fluctuates between 145 and 175 m all year round. Hence, a large number of landslide disasters have developed in the Three Gorges reservoir [3,4]. Because the Three Gorges reservoir plays an important role in flood control and power generation, it is of great significance to study geological landslides in the Three Gorges Reservoir area [5,6].

Landslide displacement prediction is a hot topic at the forefront of natural hazard research [7]. Displacement prediction is the basis of early warning systems for landslide disasters. Accurate landslide displacement prediction can reduce the losses caused by such disasters as much as possible, so as to ensure the safety of people’s lives and property. Due to the complex geological environment, the accuracy of current methods for directly predicting total displacement is not sufficient [8]. Hence, landslide displacement should be divided into several parts by the decomposition technique. At present, landslide displacement decomposition mainly adopts two methods. The first is the time series and simple moving average method [9,10]. This method is simple and practical, and the displacement component obtained has a clear physical meaning. However, due to defects of the decomposition method itself, the random displacement cannot be obtained. The second is empirical mode decomposition (EMD), wavelet analysis, and ensemble empirical mode decomposition (EEMD), which can divide the total displacement into a specific number of components, so it has clear physical significance [11,12,13].

The landslide displacement prediction model has experienced rapid development in the past 50 years, which was from the initial empirical model to the mathematical statistical model, and then to the non-linear theoretical model and the comprehensive model [14]. Nowadays, with the development of high-speed computers, various machine learning models including deep learning have been widely used for predicting landslide displacement, such as ELM (Extreme Learning Machine) [15], EML (Evaluating Machine Learning) [16], BPNN (Back Propagation Neural Network) [17,18], SVR (Support Vector Regression) [19], KELM (Kernel Extreme Learning Machine) [20,21], LSTM (Long Short-Term Memory) [9,22], and so on. Many algorithms have been used to optimize the parameters for the prediction models, including GS (Grid Search algorithm) [10], PSO (Particle Swarm Optimization) [23], GA (Genetic Algorithm) [24], FOA (Fruit Fly Optimization Algorithm) [25], GWO (Grey Wolf Optimizer) [26], and so on. Therefore, selection of the influencing factors plays a crucial role in the development of landslide prediction. Besides, for landslides in a reservoir area, the fluctuation of the reservoir level and rainfall are usually used as the hydrologic triggering factors of landslide deformation and failure [27]. However, the increase of input factors does not necessarily lead to higher prediction accuracy in the model of landslide displacement prediction. Based on the above facts, for different types of displacement, it is necessary to select the appropriate inducing factor as the input layer to establish the model. At present, data mining technology has been widely used in the field of geological hazards. Nevertheless, the research on data mining technology in landslides mostly focuses on association criterions and thresholds of triggering factors, while there are few publications on the joint use of data mining technology and deep learning. In order to optimize the triggering factors to find the most suitable factors for displacement prediction, data mining technology could be used.

In this paper, the Baishuihe landslide was taken as an example, which was in the east of Three Gorges Reservoir area. Data mining and deep learning were used for predicting the displacement. Based on the time series analysis of landslides, the displacement and triggering factors are decomposed by VMD. Periodic and random terms were predicted by FOA-BPNN. A flow chart of this work is shown in Figure 1.

## 2. Methodology

### 2.1. Two-Step Clustering

The two-step clustering algorithm is usually applied to deal with large-scale types of data, which divides and integrates data through a two-step process of pre-clustering and clustering to complete the data classification [28]. For sample data including both numerical and subtype variables, the two-step clustering algorithm usually uses a log-likelihood function. If clustered into *j* classes, it is defined as:(1)$$l=\sum_{j=1}^{J}\sum_{i\in I_{j}}\log p\left(X_{i}|\theta_{i}\right)=\sum_{j=1}^{J}l_{j}$$
where, *p* is the likelihood function; *I_j_* is the set of samples of *j*th class; *θ_j_* is the parameter vector of *j*th class; *J* is the number of clusters. For all samples, the log-likelihood clusters are obtained as the aggregation of the log-likelihood clusters for each category.

For the certain *i*th class and *j*th class, the combination is noted as <*i*, *j*>, and then their distance can be defined as:(2)$$d(i,j)=\xi_{i}+\xi_{j}-\xi_{\langle i,j\rangle }$$
where *ξ_i_* and *ξ_j_* are the log-likelihood distance of *i*th class and *j*th class, respectively. *ξ*_<*i*,*j*>_ is the log-likelihood distance of the combination of <*i*, *j*>. *ξ* is the specific form of the log-likelihood function:(3)$$\xi_{v}=-N_{V}\left(\sum_{k=1}^{K^{A}}\frac{1}{2}\log\left({\hat{\sigma }}_{k}^{2}+{\hat{\sigma }}_{vk}^{2}\right)+\sum_{k=1}^{K^{B}}E_{vk}\right)$$
where,
(4)$$\overset{\wedge }{E_{vk}}=-\sum_{l=1}^{L_{k}}\frac{N_{vkl}}{N_{v}}\log\frac{N_{vkl}}{N_{v}}$$
where *K^A^* is the number of numerical variables; *K^B^* is the number of categorical variable; ${\hat{\sigma }}_{k}^{2}$ and ${\hat{\sigma }}_{vk}^{2}$ denote the total variance of the *kth* numerical variable and the variance in *v*th class respectively; *N_v_* and *N_vkl_* are the sample size of category *v* and the first category in the *k*th subtype variable; *L_k_* is the category of the *k*th subtype variable.

After *i*th class and *j*th class are combined, −*ξ*_<*i*,*j*>_ is greater than *ξ_i_* + *ξ_j_*, and hence *d*(*i,j*) is less than 0. Moreover, the smaller *d*(*i,j*) is, the more it means that the merging of *i*th class and *j*th class will not cause a significant increase in intra-class differences. Specially, when *d*(*i,j*) is less than the threshold *C*, *i*th class and *j*th class can be merged. Conversely, when *d*(*i,j*) is greater than the threshold *C*, indicating that merging will cause a significant increase in variability within the clustered clusters, and the *i*th class and *j*th class cannot be merged.

The threshold value *C* is given by:(5)C=log(V)
(6)$$V=\prod_{k}R_{k}\prod_{m}L_{m}$$
where *R_k_* is the range of values of the *k*th numeric variable; *L_m_* is the sample size of the *m*th subtype variable.

### 2.2. Apriori Algorithm

The a priori algorithm was proposed by Agrawal [29]. This algorithm can deal only with categorical variables rather than numeric variables. The algorithm mainly includes two steps: (1) generating frequent item sets that meet the minimum support values, and (2) generating association rules that satisfy the minimum credibility in the frequent item set generated in the first step.

The frequent item set *T* contains item *a* (frequent item set). If its support is equal to or greater than the support threshold specified by the user, as shown in Equation (1), the a priori algorithm uses the iterative method of layer-by-layer searching to generate frequent item sets.
(7)$$\frac{\left|T(a)\right|}{\left|T\right|}\geq \min\;\sup p$$

Frequent *k*-item sets are used to explore and generate (*k* + 1)-item sets. The algorithm implementation process is shown in Figure 2.

Simple association rules are generated from the frequent item sets, and association rules with confidence levels greater than the threshold value are selected to form an effective rule set. If $C_{L’\to (L-L’)}$ is greater than the confidence threshold specified by the user (see Equation (2)), then the association rule can be generated.
(8)$$C_{L’\to (L-L’)}=\frac{\left|T(L)\right|}{\left|T(L’)\right|}\geq \min\;conf$$

### 2.3. VMD

Based on the EMD model, the VMD model was proposed in 2013, which is an adaptive method for signal processing and modal variation [30]. The constraint variation can be expressed as:(9)$$\left\{\begin{array}{l} \underset{\left\{u_{k}\right\}\left\{\omega_{k}\right\}}{\min}\left\{\overset{K}{\underset{k=1}{\sum }}∥\partial_{t}\left[\left(\sigma (t)+\frac{j}{\pi t}\right)*u_{k}(t)\right]e^{-j\omega_{k}t}{∥}_{2}^{2}\right\}\\ \;s.t.\;\overset{K}{\underset{k=1}{\sum }}u_{k}=f(t) \end{array}\right\}$$
where *f*(*t*) is the original signal, *K* is the number of components, $\partial_{t}$ denotes the Dirac function, $\left\{\omega_{k}\right\}$ denotes the actual central frequency, $\left\{u_{k}\right\}$ denotes the component obtained after decomposition, (σ(t)+jπt)∗uk(t) denotes the analytical signal of each component, $e^{-j\omega_{k}t}$ denotes the estimated central frequency of each analytical signal, and * denotes the convolution operator. We then obtain the following:(10)$$L\left(\left\{u_{k}\right\},\left\{\omega_{k}\right\},\lambda \right)=\alpha \underset{k}{\sum }∥\partial_{t}\left[\left(\sigma (t)+\frac{j}{\pi t}\right)*u_{k}(t)\right]e^{-j\omega_{k}t}{∥}_{2}^{2}+∥f(t)-\underset{k}{\sum }u_{k}(t){∥}_{2}^{2}+\langle \lambda (t),f(t)-\underset{k}{\sum }u_{k}(t)\rangle$$
where *λ* denotes the Lagrange multiplier.

By using the alternative direction method of multipliers (ADMM), the saddle point of the model without an upper constraint can be obtained, which is the optimal solution of the constrained variational model, so that the original signal can be decomposed into IMF components.

### 2.4. FOA-BPNN

The BPNN is a multilayer feedforward neural network based on error back propagation algorithm training, which was first proposed by Rumelhart and McClelland [31]. BPNNs have arbitrary complex pattern classification and good multidimensional function mapping ability. In addition, it can solve XOR and other problems that simple perceptrons cannot solve. Structurally, BPNNs are composed of an input layer, a hidden layer, and an output layer. The BP algorithm takes the network’s square error as the objective function and uses the gradient descent method to calculate the minimum objective function.

In addition, as proposed by Wen-Tsao Pan [32], the FOA is a new method of global optimization, which is based on the foraging behavior of *Drosophila melanogaster*. Because the fruit fly is superior to other species in terms of smell and vision, the olfactory organ of *Drosophila* can collect all kinds of smells floating in the air, even the smells of food sources 40 km away. Then, after flying to the vicinity of the food location, they can use their sharp vision to find the food or observe the gathering position of their companions, and fly in that direction. The optimization procedure of the FOA-BPNN is shown in Figure 3.

## 3. Case Study

### 3.1. Geological Settings of Three Gorges Reservoir Area

The Three Gorges reservoir is an artificial lake formed after the completion of the Three Gorges hydropower station, situated in the middle part of China. The total lengths of the Yangtze River and surrounding area are 660 km and 1084 km^2^ respectively. The altitude drops from the highest part to the west and east, forming a hilly landform and medium altitude mountains, respectively. The trend of the mountains is controlled by the main geological structures. The strata in the Three Gorges Reservoir area are from pre Sinian to Quaternary. Jurassic red strata are dominant in the Three Gorges Reservoir area, mainly exposed in the west of Zigui county east of Fengjie county (the red strata refer to sandstone, mudstone, and sandstone interbedded with mudstone layers). In addition, other sedimentary rocks (limestone, marl, and dolomite) also exist in the area between Fengjie and Zigui. These hard rocks form a steep canyon in Fengjie–Zigui area. Metamorphic complexes and magmatic rocks appear in the area near the dam site on a relatively small scale. Controlled by the complex geological conditions, coupled with seasonal rainfall and periodic fluctuation of reservoir water level, a large number of geological disasters have developed in the Three Gorges Reservoir area. A total of 4429 geological disasters have been found up to the present time, most of which are landslides, rock falls, and debris flows [4]. A geological map of the Three Gorges Reservoir area is shown in Figure 4.

### 3.2. Local Environmental Conditions

The Baishuihe landslide is in the Zigui County area of the Three Gorges Reservoir area, located in the middle latitude, belonging to a subtropical continental monsoon climate zone, with a warm and humid climate, sufficient light, abundant rainfall, and distinct seasons. The average annual rainfall of Zigui County is 1493.2 mm. Rainfall is generally concentrated in the flood season in this area, and the maximum daily rainfall has historically reached up to 358 mm. The monsoon is mainly southerly. Limited by the terrain, the wind speed is generally low. The Yangtze River is the lowest erosion base level in this area and flows through the front edge of the landslide from west to east. The cross section of the river valley is a “V” shape, steep to the north and gentle to the south. There are several gullies short in length and depth in the landslide area, all of which are trunk gullies. Only temporary flood flows are formed after rainstorms, which constitute the primary discharge channel of surface water in the area.

The Baishuihe landslide is located on the south bank (convex bank) of the Yangtze River. The elevation of the landslide gradually decreases from south to north. The elevation of the toe and rear edges is about 70 m and 400 m, and the bedrock ridge is the boundary between the eastern and western sides. The deformation of the middle and front part of the landslide is relatively strong. Pinnate fissures are continuously distributed on both sides of the boundary, and the boundary between the east side and the rear edge is basically connected. The slope of the landslide is 30° to 35°, and the landslide has an average thickness of 30 m with a volume of 1.26 × 10^7^ m^3^. The topographic map and a schematic geological profile of the Baishuihe landslide are shown in Figure 5 and Figure 6.

### 3.3. Deformation of the Landslide

The Baishuihe landslide has been monitored since June 2003, and the layout of the monitoring surface layout is shown in Figure 5b. According to the characteristics of the surface monitoring displacement and surface macro-deformations, the Baishuihe landslide can be divided into two areas; however, the active area (area *A*) is the middle and front part of the landslide, which has strong deformation. After the completion of the Three Gorges Dam, the landslide has produced obvious displacement due to the impoundment of the reservoir. Several transverse tension cracks have appeared in the east of the landslide. Specifically, the eastern and posterior boundaries are basically connected, and the western boundary’s cracks are in the shape of pinnately distributed cracks. From August 2005 to August 2006, there were many landslides on the inner side slope of the riverside highway with an elevation of about 220 m, and many subsidence and tension cracks appeared on the surface of the landslide. Approximately 100,000 m^3^ of landslide debris piled on the road towards the rear of the active area in June 2007 (Figure 7).

### 3.4. Analysis of the Monitoring Data

There are three monitoring sections and six GPS monitoring points in the active Baishuihe landslide area. Among them, the monitoring points ZG93 and ZG118 have been in place since June 2003; XD01 and XD02 were added in May 2005, and XD03, XD04 were added in October 2005. Note that the displacement of all the monitoring points is synchronous. The monitoring period of Points ZG93, ZG118, and XD01 is long, meaning that they are representative and can reflect the entire movement process of the landslide. Therefore, in this study, these three points were taken for detailed analyses (Figure 8). According to the filling scheduling of the reservoir, the monitoring data can be divided into three stages for analysis, as described below.

(1)Phase I (from June 2003 to June 2006): The water level of the reservoir started at 135 m in September and reached its highest level of 139 m in October. The maximum displacement of ZG93 and ZG118 was 25.8 mm and 30.6 mm, respectively, during the three impoundment periods. During Phase I, the reservoir basically maintained the highest water level from November to January of the next year. The maximum monthly displacement rates of these three points were below 13 mm/month during this period, which was relatively slow. The water level began to drop in February each year and reached the lowest level (135 m) in July. During this period, the minimum increase of these three points was over 80 mm, and the maximum increase was over 150 mm. Especially in May and June, the rate of increase in landslide displacement was the largest. From the end of July to the beginning of September, the reservoir water level remained at the lowest level, but the landslide displacements first continued to grow rapidly and then basically remained the same. In this stage, the water level of the Yangtze River changed from having the natural water level for many years to the manually adjusted reservoir water level, and the landslide was still in the adaptation period of adjustment of the reservoir’s water level. Therefore, we can consider that the deformation of the landslide in this stage was mainly affected by the decline in the reservoir’s water level. In particular, the heavy rainfall in July 2005 did not cause an obvious increase in the displacement of the landslide.(2)Phase II (from July 2006 to June 2008): The water level of the reservoir fluctuated between 145 and 155 m, which dropped from 155 m to 145 m for the first time during April to June 2007. Alternatively, a drastic drop in the water level led to an increase in the hydrodynamic pressure inside the landslide, which caused the displacement of each monitoring point to suddenly increase for the first time, increasing by more than 1000 mm.(3)Phase III (from July 2008 to December 2016): The water level of the reservoir fluctuated between 145 and 175 m. Before 2015, the annual displacement rate showed a downward trend.

In summary, the fluctuation in the reservoir’s water level resulted in the significant extension of the fluctuating range and immersion range of the reservoir’s water level, then, in turn, the stress field, seepage field, and rock–soil structure characteristics of the sliding mass changed significantly, which had a significant impact on the evolution process of the Baishuihe landslide. In addition, owing to the different stages of the reservoir’s water level operations, there were some differences in the degree of impact on the deformation evolution process of the landslide. Moreover, although the landslide has undergone some adjustment, its shear deformation energy has been released to a certain extent. However, when external effects such as rainfall and the reservoir water level change dramatically again, the landslide will tend to be unstable.

## 4. Results

### 4.1. Triggering Factors

Rainfall and periodic fluctuation of reservoir water level are the main inducing factors of landslide deformation [34,35]. The periodic fluctuation of the water level in the Three Gorges causes dynamic osmotic pressure in the slope, resulting in landslide deformation [36]. On the one hand, rainfall can increase the weight of the landslide mass, thus increasing the sliding force of landslides; on the other hand, it can weaken the mechanical strength of the landslide rock and soil mass, resulting in landslide deformation [37,38]. Therefore, rainfall and reservoir water level can be used as trigger factors for landslide deformation [39]. In this research, a total of 10 triggering factors were selected to carry out displacement prediction research, including 5 reservoir level related factors (monthly average water level $\bar{h}$; monthly maximum daily drop of water level $\Delta h_{\max}^{dailydrop}$; monthly maximum daily rise of water level $\Delta h_{\max}^{dailyrise}$; monthly fluctuation of water level $\Delta h_{}^{month}$; bimonthly fluctuation of water level $\Delta h_{}^{2month}$), 4 rainfall related factors (monthly maximum effective continuous rainfall $q_{continuous}^{effective}$; monthly cumulative rainfall $q_{}^{month}$; bimonthly cumulative rainfall $q_{}^{2month}$; monthly maximum daily rainfall $q_{\max}^{day}$), and 1 deformation factor (last monthly velocity of deformation *v*), as shown in Table 1. In this study, the monitoring data of ZG93 were selected for landslide prediction. In addition, because the monitoring points ZG118 and XD01 have similar deformation characteristics to ZG93, the monitoring data of ZG118 and XD01 were added to increase the sample size and overcome model overfitting errors, as well as to provide a more representative prediction of the overall landslide displacement. The triggering factors are shown in Table 1.

### 4.2. Clustering Results

In the two-step clustering algorithm, the minimum and maximum categories of the triggering factors were set as 2 and 10 respectively. In clustering algorithms, there are two commonly used clustering criteria: the Akaike Information Criterion (AIC) and the Bayesian Information Criterion (BIC). When the number of samples is large, the BIC criterion can effectively avoid the model complexity caused by high model accuracy. Therefore, in this study, the BIC was chosen as the cluster criterion, and the distance measurement method was Euclidean distance. The clustering results of the external triggering factors are shown in Table 2 and Table 3. Monthly velocity (*v*) was clustered into three categories (Low, V1; Medium, V2; High, V3), as shown in Table 4.

### 4.3. Association Rules

In the a priori algorithm, the minimum conditional support was set to 0.01 and the minimum rule confidence was set to 100% to ensure that the mining association criteria were absolutely correct. In total, 5447 association rules were generated, most of which were V1 and V2 stages (4247 and 1008, respectively). The main factors controlling V1 deformation of the landslide were smooth fluctuations of the reservoir’s water level and light rainfall. The main factors controlling V2 deformation of the landslide were sharp fluctuations of the water level and medium to heavy rainfall. The main factor controlling V3 deformation of the landslide was heavy rainfall. Nevertheless, there may be some time correlation between these nine factors. In general, a drop in reservoir water and heavy rainfall were the main factors causing landslide deformation in the Three Gorges Reservoir area. It can be seen from Figure 7 that the water level of the Three Gorges reservoir has had a period of slow decline (175 m–165 m) from January to April and a rapid decline (165 m–145 m) from April to June since 2008. The heavy rainfall is concentrated from June to September every year. Moreover, this is also a critical period when the landslide produces severe deformation.

The statistical results of the data mining and association rules are shown in Table 5. The total support, average support, and the contribution without support of each triggering factor were counted, and the comprehensive contribution was the mean value of these three contributions. The comprehensive contribution of each factor according to the association rules is shown in Figure 9. Factors with a degree of contribution less than 0.3 were eliminated and were not used as input layers in the prediction model. Therefore, eight triggering factors were taken as the input layer in the V1 and V3 prediction models (F1, F3, F5, F6, F7, F8, F9, and F10), and eight triggering factors were taken as the input layer in the V2 prediction model (F1, F2, F5, F6, F7, F8, F9, and F10).

### 4.4. Decomposition of Displacement

The non-stationary time series theory indicated that the time series consisted of three parts: the trend term, the periodic term, and the random term. For the landslide displacement, the time series can be divided into three parts: (1) trend displacement, which is controlled by internal factors, such as geological conditions, geomorphology, geological structure, rock and soil properties, etc.; (2) periodic displacement, which is controlled by external factors, such as rainfall, the reservoir’s water level, wind load, air temperature, etc.; and (3) random displacement, which is controlled by random factors, such as human activities, engineering construction, vehicle loads, vibration loads, etc., as shown in Figure 10.
(11)X(t)=α(t)+β(t)+γ(t)

Here, X(t) denotes the observed value of landslide displacement, and α(t), β(t), and γ(t) denote the trend, periodic, and random displacements, respectively.

Therefore, *K* was set to 3 and 2 in the VMD decomposition of the landslide displacements and triggering factors, respectively. The penalty parameter *a* and the rising step *τ* (*a* = 1.5 and *τ* = 0.1) were finally determined through multiple trials as follows. (1) In the displacement decomposition, *a* = 1.5 and *τ* = 0.1. (2) In the triggering factors decomposing, *a* = 700 and *τ* = 0.5. The decomposition results are shown in Table 6 and Figure 11.

### 4.5. Displacement Prediction

#### 4.5.1. Trend Term Prediction

The displacement of the trend term showed a distinct piecewise function. Therefore, the trend term of ZG93 was divided into three phases: Phase 1 (June 2003~June 2007), Phase 2 (June 2007~June 2014), and Phase 3 (June 2014~December 2016). Multiple fitting results showed that good fitting results can be obtained by using a cubic function and the robust least squares method. The fitting function can be defined as:*S* = *at*^3^ + *bt*^2^ + *ct* + *d*(12)

The fitting results and parameters are shown in Figure 12 and Table 7, which indicate that the prediction accuracy’s R^2^ and the RMSE of the trend term were 99.4% and 4.063.

#### 4.5.2. Periodic and Random Term Prediction

The periodic and random displacements were trained and predicted by FOA-BPNN. In general, a model’s performance is usually affected by its own structure. Through extensive sensitivity analysis, the most reliable structure can be obtained [40,41]. Therefore, in the process of FOA optimization, 12 different structures with population sizes between 10 and 120 (10 intervals) were tested [42,43]. Each network was executed with 100 repetitions, and the MSE (between the actual and predicted periodic displacements of the landslide) was defined as the objective function used to evaluate the performance error of the model. It is worth noting that each structure was tested five times to evaluate its repeatability. The sensitivity curves are shown in Figure 13, which indicate that the MSE of the model decreases with an increase in the population size. However, because FOA-BPNN integration reduces the error in the training process, the model is less sensitive to population size. The computing time of models with different population sizes is shown in Figure 14. After consideration of the calculation costs and error, the population size of FOA-BPNN model was determined as 10. In total, six FOA-BPNN models were built, including individual periodic prediction models for V1, V2, and V3, and individual random prediction models for V1, V2, and V3, as shown in Figure 15.

Figure 15 indicates that the proposed models achieved good prediction results. According to the results of the residual error analysis, in the training process of the model, the residual error of displacement was relatively stable, which also verified the robustness and reliability of the model. For the prediction samples, there were some fluctuations in the residual error. The prediction accuracy of the model will be analyzed in the Discussion section.

#### 4.5.3. Total Displacement

The total displacement prediction results of the landslide can be obtained by superimposing the prediction results of all three types of displacement, as shown in Figure 16, which shows that the prediction model achieved good accuracy for monitoring point ZG93. In June 2007, there was a significant difference between the total displacement training value and the actual value, resulting in the obvious mutation of the residual error. This was because in the three parts of landslide displacement (trend, periodic, random), the trend displacement accounts for more than 85%. In June 2007, it was the boundary between Phase 1 and Phase 2, where there were some differences in the training results of the two polynomial fitting functions, resulting in a large residual error in the total displacement. However, the residual error was relatively stable for the prediction samples.

## 5. Discussion

As mentioned above, the landslide displacement contains three parts: (1) trend displacement, which is controlled by internal factors; (2) periodic displacement, which is controlled by external factors; and (3) random displacement, which is controlled by random factors. Generally, in predictions of landslide displacement, the selection of triggering factors is based on monitoring data such as rainfall and the reservoir’s water level, which are the main factors causing periodic displacement. Therefore, taking these factors as the input of periodic displacement will not only have clear physical significance but will also significantly improve the accuracy of landslide displacement predictions. When the time series analysis method was used to predict the landslide displacement, the displacement trend was relatively easy to predict. Therefore, choosing the appropriate periodic displacement prediction model is the key to improving the effect of landslide displacement predictions. Moreover, landslide prediction models have experienced rapid development in the past 50 years, and various machine learning models have been widely used for predicting landslide displacements. However, each algorithm has its limitations. For instance, SVM has low computational complexity but it is sensitive to the choice of parameters and kernel function. The decision tree model does not need any prior assumptions on the data, but the required sample size is relatively large, and its ability to deal with missing values is quite limited. ELM uses the principle of least squares and a pseudo-inverse matrix to solve the problem, which is only suitable for single-hidden-layer neural networks. BPNN has strong self-learning, self-adaptive ability, and good generalization ability but it is prone to slow convergence. In this study, based on the VMD and data mining results, the FOA-BPNN was used to predict the periodic and random terms of monitoring point ZG93′s displacement. The BPNN, SVM, and ELM algorithms were chosen as the comparison models (Models 2–4). The performance of various displacement prediction models of the Baishuihe landslide are shown in Table 8 and Figure 17. The prediction accuracy of the FOA-BPNN model was the highest. The R^2^ reached 0.977 and its RMSE was only 10.041. In contrast, the proposed model could improve the accuracy of landslide displacement predictions.

In this study, ZG93’s monitoring data were selected for predicting displacement, and the monitoring data of points ZG118 and XD01 were added to increase the sample size and overcome the model’s overfitting error, and to provide a better representative prediction of the overall landslide displacement. The accuracy of various models in terms of predicting ZG93’s displacement has been discussed. The monitoring points ZG118 and XD01 in 2016 were used for the model validation. The measured and predicted displacements of ZG93, ZG118, XD01 are shown in Figure 18. The R^2^ values between the measured and predicted displacements of ZG118 and XD01 were 0.977 and 0.978, respectively. The RMSE of these two monitoring points was 12.40 and 16.04, respectively. In the previous study [10], cumulative displacement was divided into trend term and periodic term by time series model and moving average method. A cubic polynomial model was proposed to predict the trend term of displacement. Then, multiple algorithms were used to determine the optimal support vector regression (SVR) model and train and predict the periodic term. In this paper, data mining technology is used to screen the trigger factors of periodic items, and the more advanced FOA optimization algorithm is used to optimize the parameters of the machine learning model. Furthermore, this paper uses a VMD model to divide the landslide displacement data, which makes great progress compared with the moving average model, and will be more conducive to the integration and automation of the landslide prediction model. Therefore, the prediction accuracy obtained in this paper (R^2^ = 0.977 and 0.978) is significantly higher than that of previous studies (R^2^ = 0.963 and 0.951). In general, the model proposed in this study has achieved good results in terms of predicting the displacement of different monitoring points of the landslide, which has high practicability and application value in the study of landslide displacement predictions. However, it is worth noting that due to the small amount of displacement data in the V3 state of the monitoring point (Table 6), the prediction results of XD01 have obvious errors for July 2016. Therefore, in order to obtain satisfactory prediction results, the monitoring data of various states should be supplemented as much as possible.

## 6. Conclusions

In this paper, the Baishuihe landslide in the Three Gorges Reservoir area was taken as an example. Data mining and deep learning were used for displacement prediction. The following conclusions can be reached:(1)Using VMD to decompose the displacement of Baishuihe landslide can correspond to the triggering factors, which had clear physical significance.(2)The association rules showed that the main factors controlling the V2 and V3 deformation of the landslide were the sharp fluctuation of reservoir water level and medium–heavy rainfall.(3)R^2^ between the measured and prediction displacements of ZG118 and XD01 were 0.977 and 0.978. RMSE of these two monitoring points were 12.40 and 16.04, respectively.(4)An integrated approach for landslide displacement prediction including data mining and deep learning was proposed, which could guide the managers of geological disasters to improve the prediction accuracy, so as to reduce the losses caused by landslides.

## Figures and Tables

**Figure 1 sensors-22-00481-f001:**
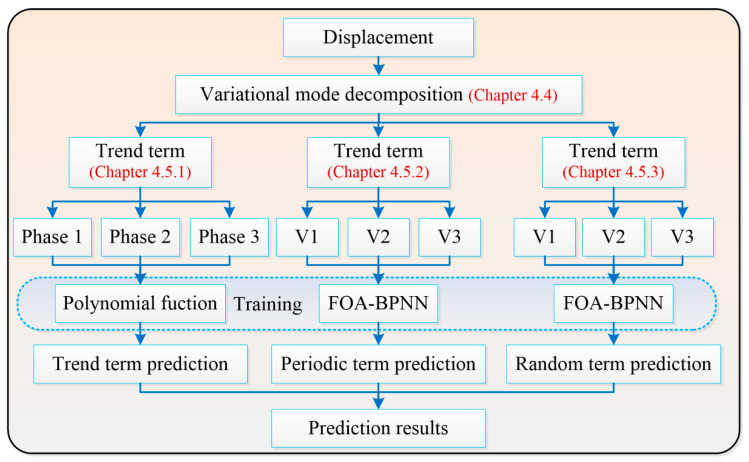
Flow chart of the displacement prediction.

**Figure 2 sensors-22-00481-f002:**
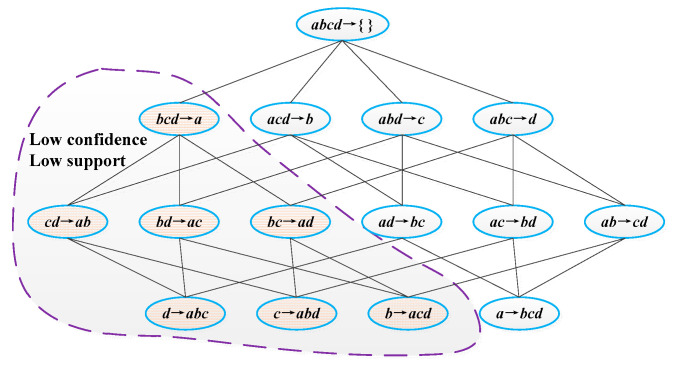
The implementation process of the a priori algorithm.

**Figure 3 sensors-22-00481-f003:**
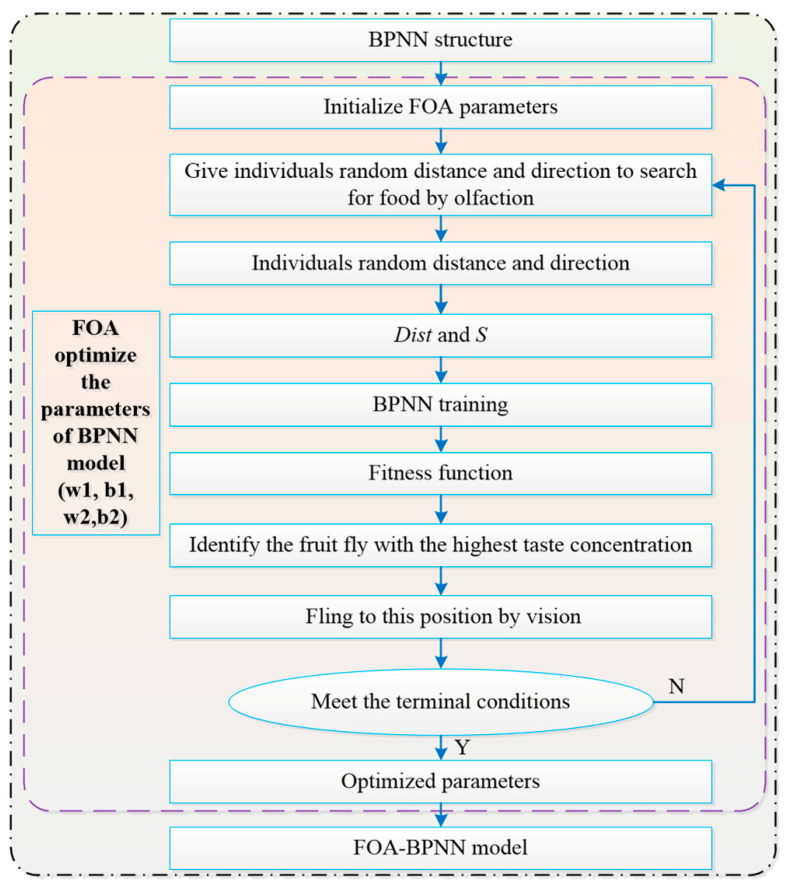
Optimization procedure of an FOA-BPNN.

**Figure 4 sensors-22-00481-f004:**
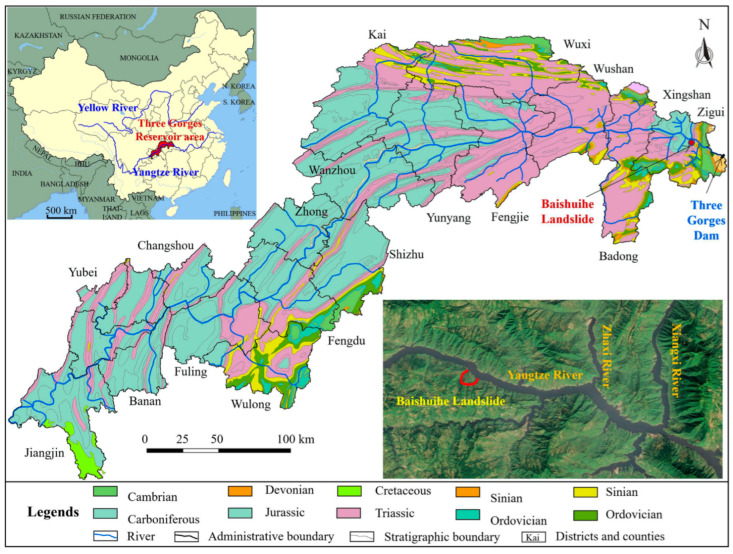
Geological map of the Three Gorges Reservoir area.

**Figure 5 sensors-22-00481-f005:**
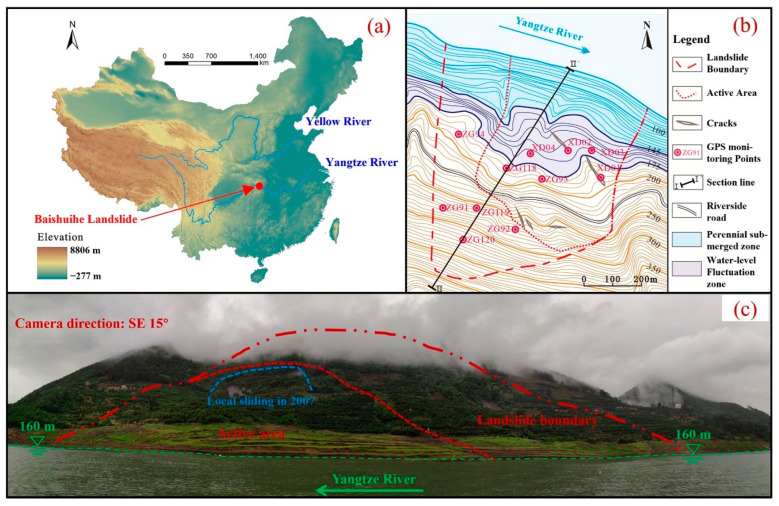
(**a**) Location of the Baishuihe landslide; (**b**) Topographic map of the Baishuihe landslide; (**c**) Overall view of the Baishuihe landslide © 2022 Springer Nature [33].

**Figure 6 sensors-22-00481-f006:**
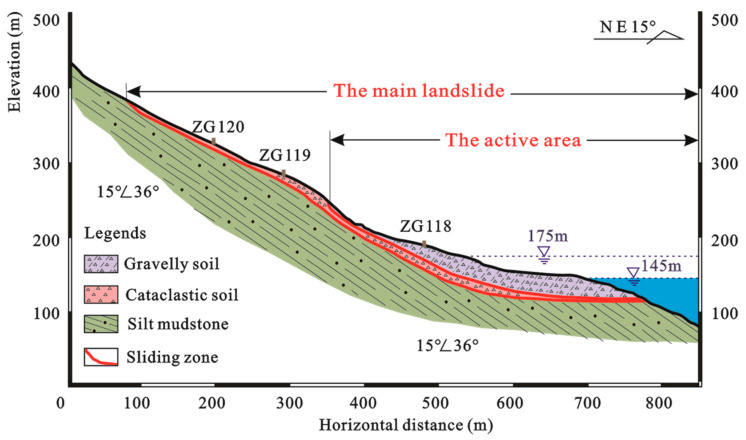
Schematic geological profile of the Baishuihe landslide (II–II’) © 2022 Springer Nature [10].

**Figure 7 sensors-22-00481-f007:**
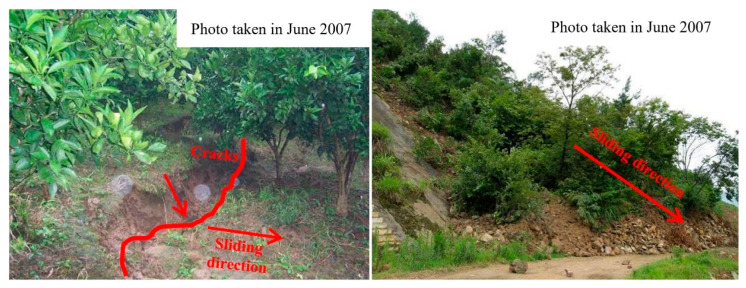
Macroscopic deformation of the Baishuihe landslide © 2022 Springer Nature [33].

**Figure 8 sensors-22-00481-f008:**
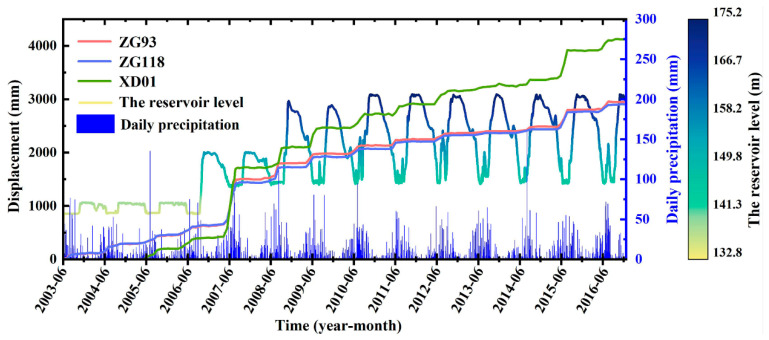
Long term monitoring data of the Baishuihe landslide (displacement, reservoir level, precipitation).

**Figure 9 sensors-22-00481-f009:**
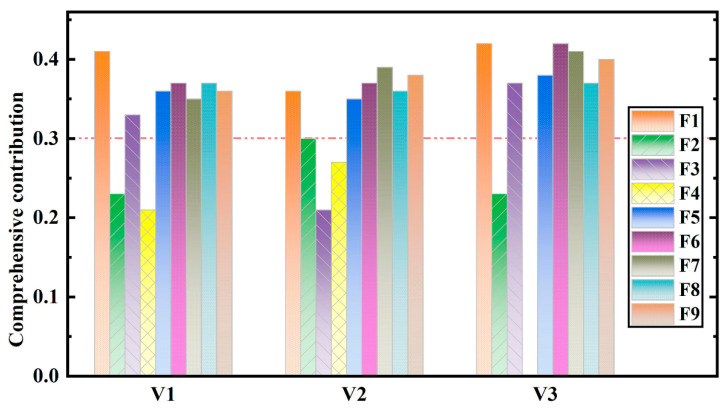
Comprehensive contribution of each factor according to the association rules (V1: Low monthly velocity; V2: Medium monthly velocity; V3: High monthly velocity; F1: Monthly average water level; F2: Maximum monthly daily drop in water level; F3: Maximum monthly daily rise in water level; F4: Monthly fluctuation of the water level; F5: Bimonthly fluctuation of the water level; F6: Maximum monthly effective continuous rainfall; F7: Cumulative monthly rainfall; F8: Cumulative bimonthly rainfall; F9: Maximum monthly daily rainfall).

**Figure 10 sensors-22-00481-f010:**
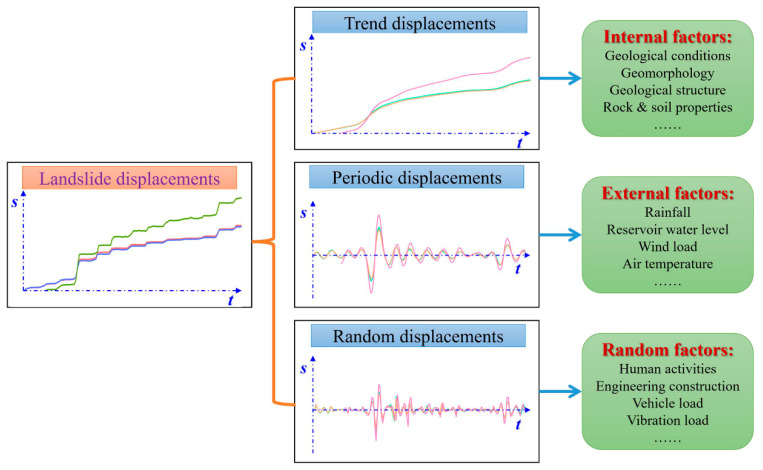
Relationship among trend displacement, periodic displacement, and random displacement.

**Figure 11 sensors-22-00481-f011:**
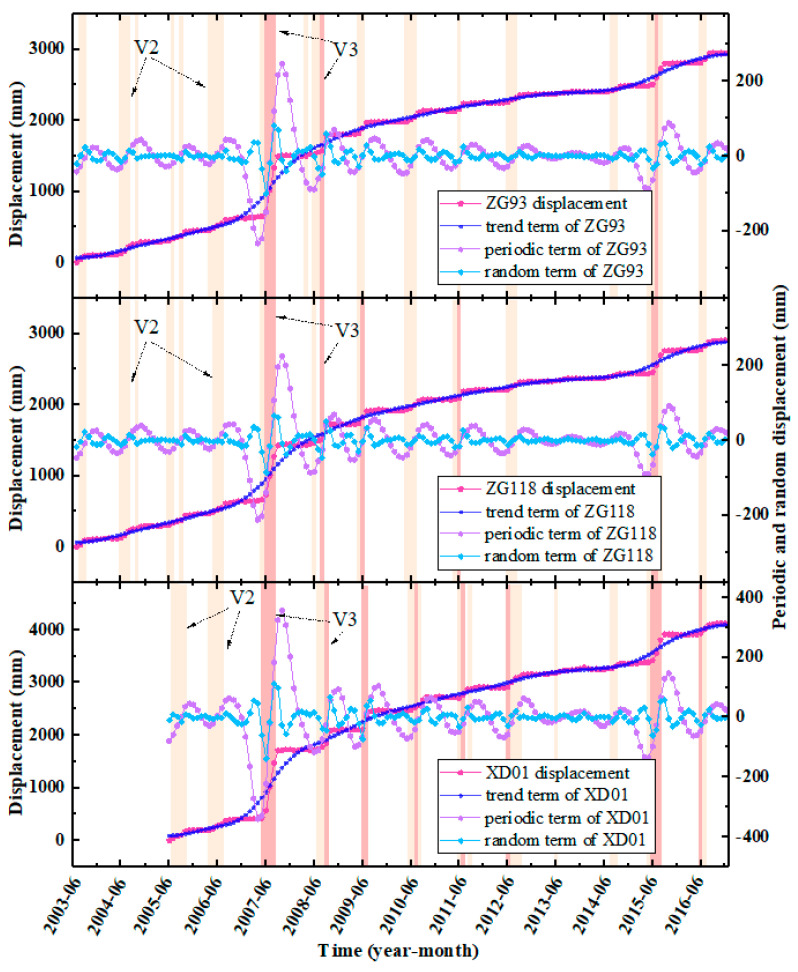
VMD decomposition of displacements at the monitoring points (ZG93, ZG118, XD01).

**Figure 12 sensors-22-00481-f012:**
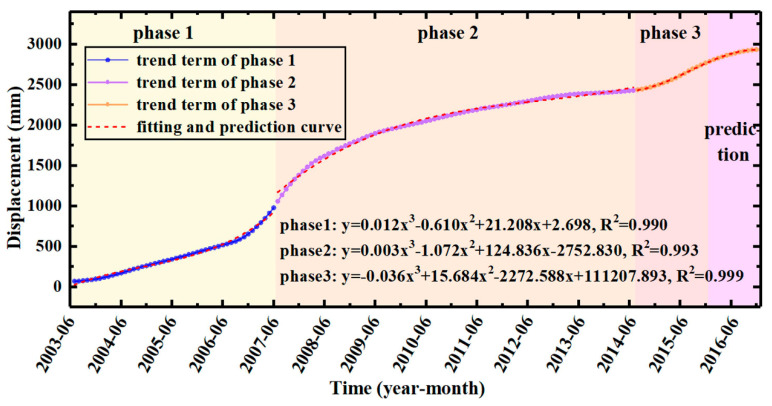
Fitting and prediction curves of the trend term.

**Figure 13 sensors-22-00481-f013:**
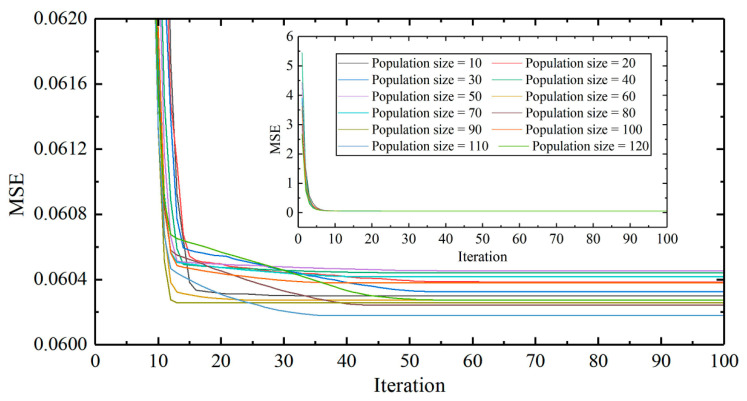
A population-based sensitivity analysis for the FOA-BPNN model.

**Figure 14 sensors-22-00481-f014:**
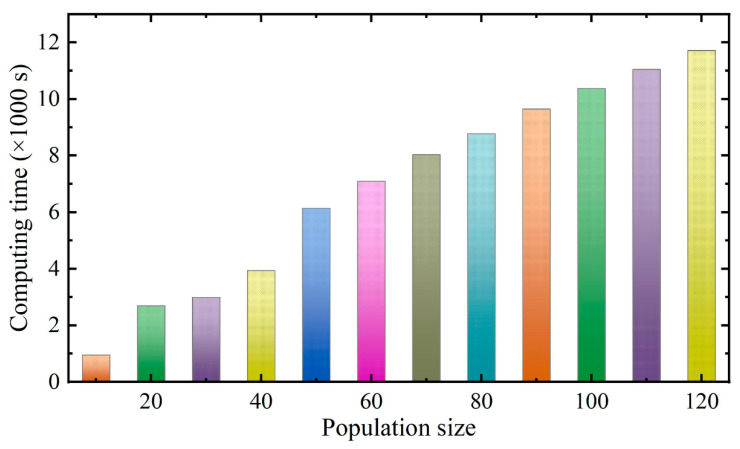
Computing time of different population sizes in MATLAB2019 software.

**Figure 15 sensors-22-00481-f015:**
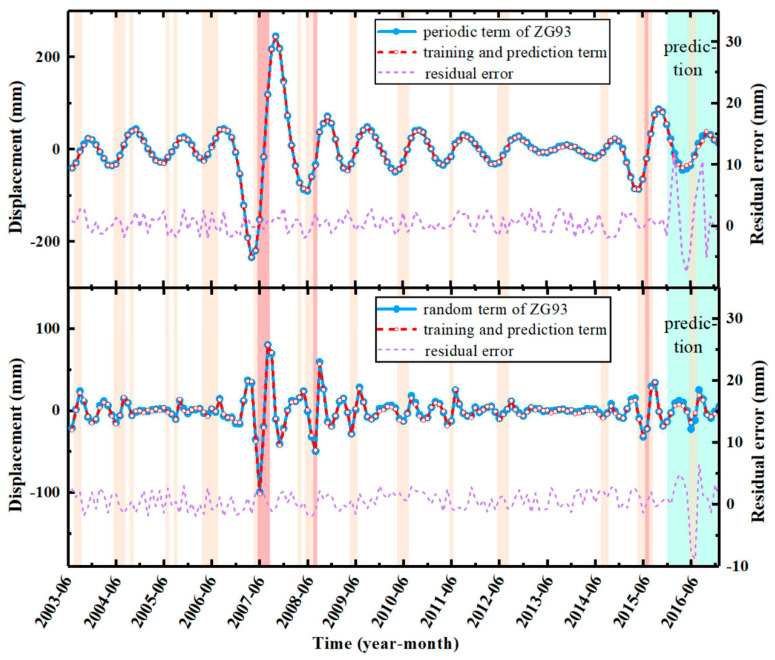
Training and prediction curves of the periodic and random terms.

**Figure 16 sensors-22-00481-f016:**
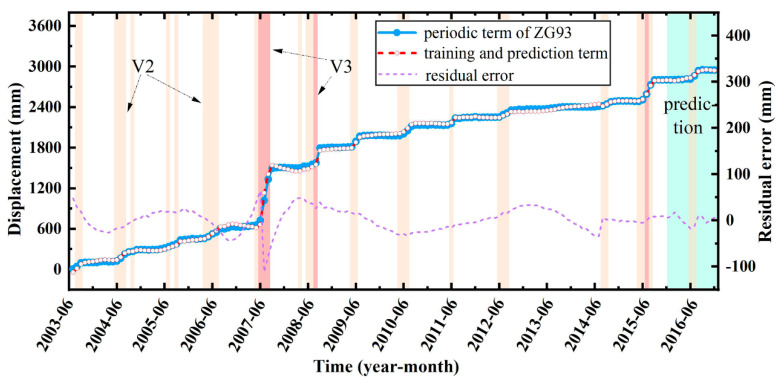
Training and prediction curves of the total displacement.

**Figure 17 sensors-22-00481-f017:**
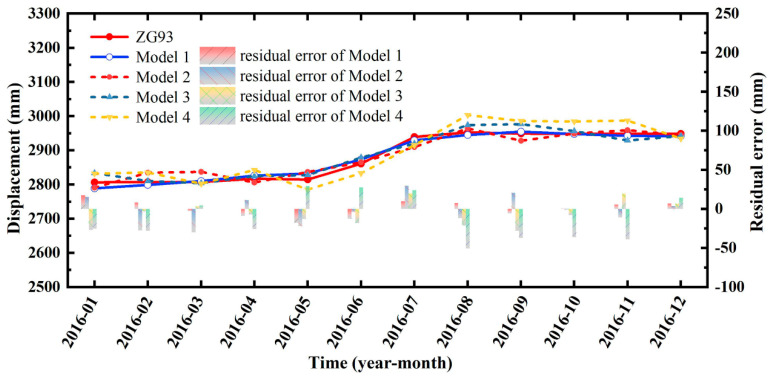
Prediction curves of the total displacement.

**Figure 18 sensors-22-00481-f018:**
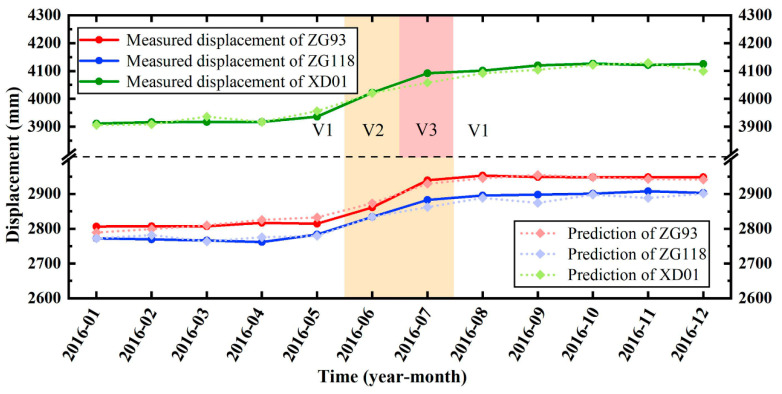
Measured and predicted displacements of ZG93, ZG118, and XD01.

**Table 1 sensors-22-00481-t001:** Triggering factors used to carry out displacement predictions.

No.	Factors	Category
F1	$\mathrm{Monthly}\;\mathrm{average}\;\mathrm{water}\;\mathrm{level}\;(\bar{h})$ (m)	Reservoir water
F2	$\mathrm{Maximum}\;\mathrm{monthly}\;\mathrm{daily}\;\mathrm{drop}\;\mathrm{in}\;\mathrm{water}\;\mathrm{level}\;(\Delta h_{\max}^{dailydrop})$ (m/day)	Reservoir water
F3	$\mathrm{Maximum}\;\mathrm{monthly}\;\mathrm{daily}\;\mathrm{rise}\;\mathrm{in}\;\mathrm{water}\;\mathrm{level}\;(\Delta h_{\max}^{dailyrise})$ (m/day)	Reservoir water
F4	$\mathrm{Monthly}\;\mathrm{fluctuation}\;\mathrm{of}\;\mathrm{the}\;\mathrm{water}\;\mathrm{level}\;(\Delta h_{}^{month})$ (m/month)	Reservoir water
F5	$\mathrm{Bimonthly}\;\mathrm{fluctuation}\;\mathrm{of}\;\mathrm{the}\;\mathrm{water}\;\mathrm{level}\;(\Delta h_{}^{2month})$ (m/2 months)	Reservoir water
F6	$\mathrm{Maximum}\;\mathrm{monthly}\;\mathrm{effective}\;\mathrm{continuous}\;\mathrm{rainfall}\;(q_{continuous}^{effective})$ (mm)	Rainfall
F7	$\mathrm{Cumulative}\;\mathrm{monthly}\;\mathrm{rainfall}\;(q_{}^{month})$ (mm)	Rainfall
F8	$\mathrm{Cumulative}\;\mathrm{bimonthly}\;\mathrm{rainfall}\;(q_{}^{2month})$ (mm)	Rainfall
F9	$\mathrm{Maximum}\;\mathrm{monthly}\;\mathrm{daily}\;\mathrm{rainfall}\;(q_{\max}^{day})$ (mm)	Rainfall
F10	Monthly velocity (*v*) (mm/month)	Deformation

**Table 2 sensors-22-00481-t002:** Clustering results of the reservoir water level factors (ZG93, ZG118, XD01).

No.	Factors	Clustering Results		Count
F1	$\bar{h}$	(135.13~138.95)	High Water Level (F11)	97
		(144.21~158.02)	Medium Water Level (F12)	186
		(160.14~174.74)	Low Water Level (F13)	183
F2	$\Delta h_{\max}^{dailydrop}$	(−0.14~0.58)	Slow Daily Drop (F21)	339
		(0.63~1.87)	Medium Daily Drop (F22)	92
		(1.91~3.69)	Sharp Daily Drop (F23)	35
F3	$\Delta h_{\max}^{dailyrise}$	(−0.43~0.04)	Slow Daily Rise (F31)	129
		(−1.70~−0.49)	Sharp Daily Rise (F32)	337
F4	$\Delta h_{}^{month}$	(0~6.18)	Smooth Fluctuation (F41)	349
		(6.59~18.25)	Sharp Fluctuation (F42)	117
F5	$\Delta h_{}^{2month}$	(0~6.50)	Non-fluctuation (F51)	250
		(6.68~14.15)	Smooth Fluctuation (F52)	126
		(14.91~28.71)	Sharp Fluctuation (F53)	90

**Table 3 sensors-22-00481-t003:** Clustering results of the rainfall factors (ZG93, ZG118, XD01).

No.	Factors	Clustering Results		Count
F6	$q_{continuous}^{effective}$	(1.50~30.30)	Light Effective Rainfall (F61)	182
		(31.30~66.00)	Moderate Effective Rainfall (F62)	151
		(67.70~110.50)	Medium Effective Rainfall (F63)	92
		(125.00~239.40)	Heavy Effective Rainfall (F64)	41
F7	$q_{}^{month}$	(3.10~66.10)	Light Effective Rainfall (F71)	198
		(69.90~163.70)	Moderate Effective Rainfall (F72)	191
		(168.50~291.50)	Medium Effective Rainfall (F73)	60
		(357.50~517.60)	Heavy Effective Rainfall (F74)	17
F8	$q_{}^{2month}$	(18.40~135.20)	Light Effective Rainfall (F81)	197
		(143.60~362.90)	Moderate Effective Rainfall (F82)	212
		(367.20~726.30)	Heavy Effective Rainfall (F83)	57
F9	$q_{\max}^{day}$	(1.30~25.60)	Light Daily Rainfall (F91)	234
		(26.50~51.30)	Moderate Daily Rainfall (F92)	151

**Table 4 sensors-22-00481-t004:** Clustering results of the monthly velocity (ZG93, ZG118, XD01).

Monthly Velocity (*v*) (mm/month)	Clustering Results	Count
(−9.61~21.66)	Low (V1)	358
(22.35~81.89)	Medium (V2)	81
(137.70~313.24)	High (V3)	27

**Table 5 sensors-22-00481-t005:** Statistical results of the data mining and association rules.

Contribution		F1	F2	F3	F4	F5	F6	F7	F8	F9
V1	Association rules	2860	1936	2071	1683	2673	2780	2610	2770	2630
	Total support	4480.98	1867.49	3231.69	1557.06	3723.31	3776.06	3579.76	3800.01	3744.18
	Average support	1.57	0.96	1.56	0.93	1.39	1.36	1.37	1.37	1.42
	Contribution without support	0.67	0.45	0.49	0.40	0.63	0.65	0.61	0.65	0.62
	Comprehensive contribution	0.41	0.23	0.33	0.21	0.36	0.37	0.35	0.37	0.36
V2	Association rules	632	463	308	392	630	654	694	628	725
	Total support	453.26	344.78	195.09	289.57	438.03	467.48	506.75	447.84	478.52
	Average support	0.72	0.74	0.63	0.74	0.69	0.71	0.73	0.71	0.66
	Contribution without support	0.63	0.46	0.31	0.39	0.63	0.65	0.69	0.62	0.72
	Comprehensive contribution	0.36	0.30	0.21	0.27	0.35	0.37	0.39	0.36	0.38
V3	Association rules	130	48	109	0	111	133	126	105	124
	Total support	83.43	29.45	69.32	0	70.54	81.59	80.98	67.26	76.07
	Average support	0.64	0.61	0.64	0	0.64	0.61	0.64	0.64	0.61
	Contribution without support	0.71	0.26	0.60	0	0.61	0.73	0.69	0.58	0.68
	Comprehensive contribution	0.42	0.23	0.37	0	0.38	0.42	0.41	0.37	0.40

**Table 6 sensors-22-00481-t006:** Composition of training and prediction samples.

Samples	Training Samples			Prediction Samples		
Monthly velocity	V1	V2	V3	V1	V2	V3
ZG93	116	30	5	10	2	0
ZG118	119	24	8	10	2	0
XD01	93	22	13	10	1	1
Total samples	328	76	26	30	5	1

**Table 7 sensors-22-00481-t007:** Parameters of the trend term of displacement based on polynomial fitting.

Phase	a	b	c	d	R^2^	MSE	RMSE
Phase 1	0.012	−0.610	21.208	2.698	0.990	518.271	22.766
Phase 2	0.003	−1.072	124.836	−2752.830	0.993	780.995	27.946
Phase 3	−0.036	15.684	−2272.588	111,207.893	0.999	20.876	4.569
All training samples	/	/	/	/	0.994	563.729	23.743
Prediction samples	/	/	/	/	0.991	16.510	4.063

**Table 8 sensors-22-00481-t008:** Performance of various displacement prediction models of the Baishuihe landslide.

Model	Algorithm’s Combination	Prediction Term		
		R^2^	MSE	RMSE
Model 1	VMD + FOA-BPNN	0.977	100.828	10.041
Model 2	VMD + BPNN	0.923	340.481	18.452
Model 3	VMD + SVM	0.944	282.566	16.81
Model 4	VMD + ELM	0.877	940.462	30.667

## Data Availability

Not applicable.

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
