# Peer review of "Data Mining and Deep Learning for Predicting the Displacement of “Step-like” Landslides"

_sensors, 2022, doi:10.3390/s22020481_

Round 1

Reviewer 1 Report

In this study, the landslide displacement of Baishuihe landslide has been predict using FOA-BPNN and VMD methods. However, there are some serious problems should be solved, which I believe that they are more important than these. They are summarized as follows:

  • In the former sentences of Abstract, the main question to be solved in this paper has not been described clearly. The main purpose in this study is to predict landslide displacement, so I think this should be described clearly.
  • In Introduction, it is not well written and needed to be improved. In the first paragraph, I think the sentence of “Landslide displacement prediction is one of the engineering geological problems that have not been solved at present, especially for the mountain and reservoir area” should move to somewhere. And it is confused to “landslide prediction” and “landslide displacement prediction”.

The reason, significance and research status for landslide displacement prediction should be described clearly.

There is the research status about machine learning models, however, the method in this study is using deep method. Why to use FOA method in this study?

  • Figure1 shows the flowchart of the displacement prediction in this study, however, it seems to be very complex. It is suggested to have a description using a chapter.
  • In Table 1, there are some triggering factors to carry out displacement prediction, however, there is no detail description about the reasons to select those factors and how to obtain those factors. Moreover, I don’t know those triggering factors are Results or basic data?
  • It can be found that there are lots of contents about the Methods in the Results. It is better to move them to the part of “Methodology”. For example, the results show that the clustering method has been used in the research, however, this method has not been introduced in the chapter of “Methodology”.
  • In Discussion, in the sentence of “Therefore, the prediction accuracy obtained in this paper (R2=0.977&0.978) is significantly higher than that of previous studies (R2=0.963&0.951)”. I doubt this. Please have a detailed explanation.
  • In the part of Conclusion, there are some conclusions that are known before this study, hence, it is suggested to draw the conclusions that revealed by this research.
  • c. should be redrawn in order to get a better effort of illustration. The writing ability of the paper needs further improvement.

Author Response

Thank you for your letter and for the reviewers’ comments concerning our manuscript entitled “Data mining and deep learning for displacement prediction of “step-like” landslide” (sensors-1510990). Those comments are all valuable and very helpful for revising and improving our paper, as well as the important guiding significance to our researches. We have studied comments carefully and have made correction which we hope to meet with approval. Revised portion are marked in red in the paper. The main corrections in the paper and the responds to the reviewer’s comments are shown in "Response to Reviewer#1.

We tried our best to improve the manuscript and made some changes in the manuscript. And here we have marked in red in the revised paper. We appreciate for Editors/Reviewers’ warm work earnestly and hope that the correction will meet with approval.

Once again, thank you very much for your good comments and suggestions.

Sincerely yours,

Yiping Wu

Reviewer 2 Report

I have reviewed the geological aspects of the study, and I need to say I miss the geological setting section in this paper, where you can describe the geological conditions of the reservoir area and the specific landslides, as well as hydrological and meteorological conditions. See my detailed comments attached. 

Author Response

Thank you for your letter and for the reviewers’ comments concerning our manuscript entitled “Data mining and deep learning for displacement prediction of “step-like” landslide” (sensors-1510990). Those comments are all valuable and very helpful for revising and improving our paper, as well as the important guiding significance to our researches. We have studied comments carefully and have made correction which we hope to meet with approval. Revised portion are marked in red in the paper. The main corrections in the paper and the responds to the reviewer’s comments are shown in "Response to Reviewer#2".

We tried our best to improve the manuscript and made some changes in the manuscript. And here we have marked in red in the revised paper. We appreciate for Editors/Reviewers’ warm work earnestly and hope that the correction will meet with approval.

Once again, thank you very much for your good comments and suggestions.

Sincerely yours,

Yiping Wu

Round 2

Reviewer 1 Report

The author has revised the manuscript and responded clearly. I recommend that the paper can be published.

Author Response

Thank you so much for your good comments.

Reviewer 2 Report

The authors have done a good job to improve the manuscript. A few additional minor comments are in the attached pdf file. 

Author Response

Thank you for your good comments. Those comments are all valuable and very helpful for revising and improving our paper, as well as the important guiding significance to our researches. We have studied comments carefully and have made correction which we hope to meet with approval. Revised portion are marked in red in the paper. The main corrections in the paper and the responds to the reviewer’s comments are shown in "Response to Reviewer#2-2".
